# Exploring the Performance of an Artificial Intelligence-Based Load Sensor for Total Knee Replacements

**DOI:** 10.3390/s24020585

**Published:** 2024-01-17

**Authors:** Samira Al-Nasser, Siamak Noroozi, Adrian Harvey, Navid Aslani, Roya Haratian

**Affiliations:** 1Bournemouth University, Fern Barrow, Poole BH12 3BB, UK; salnasser@bournemouth.ac.uk (S.A.-N.); rharatian@bournemouth.ac.uk (R.H.); 2Royal Bournemouth Hospital, Castle Ln E, Bournemouth BH7 7DW, UK; arharvey@doctors.net.uk; 3Innovid Tech Ltd., 107b Athenlay Road, London SE15 3EJ, UK; navid.aslani3600@gmail.com

**Keywords:** joint force sensor, intraoperative load measuring, artificial intelligence, total knee replacement

## Abstract

Using tibial sensors in total knee replacements (TKRs) can enhance patient outcomes and reduce early revision surgeries, benefitting hospitals, the National Health Services (NHS), stakeholders, biomedical companies, surgeons, and patients. Having a sensor that is accurate, precise (over the whole surface), and includes a wide range of loads is important to the success of joint force tracking. This research aims to investigate the accuracy of a novel intraoperative load sensor for use in TKRs. This research used a self-developed load sensor and artificial intelligence (AI). The sensor is compatible with Zimmer’s Persona Knee System and adaptable to other knee systems. Accuracy and precision were assessed, comparing medial/lateral compartments inside/outside the sensing area and below/within the training load range. Five points were tested on both sides (medial and lateral), inside and outside of the sensing region, and with a range of loads. The average accuracy of the sensor was 83.41% and 84.63% for the load and location predictions, respectively. The highest accuracy, 99.20%, was recorded from inside the sensing area within the training load values, suggesting that expanding the training load range could enhance overall accuracy. The main outcomes were that (1) the load and location predictions were similar in accuracy and precision (*p* > 0.05) in both compartments, (2) the accuracy and precision of both predictions inside versus outside of the triangular sensing area were comparable (*p* > 0.05), and (3) there was a significant difference in the accuracy of load and location predictions (*p* < 0.05) when the load applied was below the training loading range. The intraoperative load sensor demonstrated good accuracy and precision over the whole surface and over a wide range of load values. Minor improvements to the software could greatly improve the results of the sensor. Having a reliable and robust sensor could greatly improve advancements in all joint surgeries.

## 1. Introduction

The device made and used in this paper was an intraoperative load sensor that used artificial intelligence (AI) to predict the load and location of force in the knee during surgery. This device temporarily replaced the tibial insert to predict the load in both the medial (inside) and lateral (outside) compartments of the knee to allow surgeons to balance the soft tissue in real time with the use of an objective measuring tool. The introduction described the need for such a device by explaining the total knee replacement (TKR) procedure and failure and current devices and their limitations.

### 1.1. Total Knee Replacements

A total knee replacement (TKR) is the best option for pain relief and restoring function to the knee joint in patients with arthritis. The only treatment for end-stage osteoarthritis (OA) is a knee replacement due to irreparable damage to the articular cartilage [1]. Knee replacement surgery is needed when the knee joint is so worn or damaged that there is reduced mobility and increased pain. There are several factors that contribute to the increase in knee replacement surgeries in the United Kingdom (UK), including the ageing population, longer life expectancy, and an increase in BMI. Over the last three years in the UK (1 January 2018 to 31 December 2020), 226,350 primary TKRs were performed [2]. According to the National Joint Registry (NJR), TKRs are performed mostly for end-stage OA in the UK, where in the NJR’s 18th annual report, OA was listed as the reason for surgery in 96.6% of all primary knee procedures [2]. However, due to the complicated nature of knee joints, including forces and joint tension, there can be a need for early revision surgeries. According to the NJR, knee replacement surgeries have been performed in the UK since the 1970s, and about 6% of the surgeries require revision [2]. Revision TKRs are complex procedures and carry greater risks to the patients and higher costs for the NHS than primary TKRs. According to the NHS, the average cost for a revision surgery is about GBP 20,000 [3], and over 10 years, revision knee surgeries have cost the NHS over 1.7 billion pounds [2].

Reducing the number of early revision TKRs would be of benefit to the NHS and the patients. To do this, soft tissue balancing in the joint should be ensured. Ample research supports this idea where postoperative instability was reported as a major cause for early TKR revisions [4], and more research observed that 50% of early revision TKRs were related to instability, malalignment, or fixation problems, confirming imbalance as a significant cause for revision [5]. Another study found that unbalanced ligaments accounted for 35% of early TKR revisions [6]. Through a comparison of the revision rates in TKRs, it was concluded that the theoretical advantage of having a well-balanced knee is an improvement in the implant’s longevity [7]. Functional and patient-reported outcomes were also studied to confirm their improvement when the knee is balanced. Notably, patients with balanced knees are more likely to have an increased ROM and decreased pain [8]. Gustke et al. reported that balanced knees have better pain, functional, and activity scores than unbalanced knees after a one-year follow-up [9].

However, currently, balancing the knee is performed by the surgeon holding the leg and ‘feeling’ for a balanced knee. This artisan technique of joint balancing does not always produce a balanced knee. Research by MacDessi et al. demonstrated that surgeons struggle to identify a balanced knee where in an analysis of 322 TKRs, expert surgeons were only able to accurately determine a balanced knee 63% of the time at 10° of flexion, 57.5% at 45° of flexion, and 63.7% at 90° of flexion [10,11]. Therefore, there is a need for an accurate joint force measuring device to allow for the balancing of the joint intraoperatively.

### 1.2. Current Devices and Limitations

Currently, there are two intraoperative sensors that have been researched, which are VERASENSE (Orthosensor) and eLibra (Zimmer Biomet). One study found that TKRs balanced with VERASENSE had lower medial and lateral compartmental loads than manually balanced knees, concluding that sensors provide objective feedback for soft tissue balancing and can potentially improve knee balancing and rotational alignment [12]. Reference [13] compared sensor-assisted TKRs to manually balanced TKRs and found that the sensor-assisted group had improved ligament balancing with a significant reduction in manipulation under anaesthesia required after the TKR. Another study compared 75 intraoperative sensor-guided TKRs with a control group in which balancing was obtained using classic instruments and found that the sensor group had a substantially lower unexpected usage of constrained inserts (5.3%) when compared to the control group (13.8%) [14]. Constrained inserts limit varus–valgus and rotational movement [15] and lead to increased polyethylene wear, which leads to osteolysis and component loosening [16,17,18,19]. A systematic review compared manually balanced knees to sensor-balanced knees and, although it did not find a statistically significant reduction in ROM or functional outcomes, found that there was a reduction in manipulation under anaesthesia [20]. Although there have been good results of knees balanced with VERASENSE, there is a need for a more robust and comprehensive sensor because of the limitations of both devices.

First, for both VERASENSE and eLibra, the sensors were in a triangle shape in the medial and lateral compartments, meaning the sensing area is only within this area. Therefore, when contact points were outside, the accuracy of the device was greatly compromised. A study on VERASENSE’s accuracy confirmed this by finding that the biases for loading in the areas outside of the sensing area were more than two (outer anterior) and three (outer posterior) times greater than that of loading within the sensing area [21]. Additionally, VERASENSE was only calibrated to withstand passive forces, up to 310 N (69.7 lbf) and eLibra 300 N (67.4 lbf) in each compartment, and therefore, when research was being carried out on active loading, the devices were no longer reliable [22].

## 2. Self-Developed Load Sensor

### 2.1. Housing Unit

The load transducer used in this research included a metal housing unit with slits for load propagation to the sensing units. The housing unit of the transducer uses the Persona Knee System’s complex adjustable shim design to increase the thickness while using the sensor. Specifically, the transducer in this research was made to fit the Persona Knee System by Zimmer Biomet in size E/F for the right knee. The sensor replaced the Persona TASP Top Right CR 3-11/EF part and was made from an aluminium alloy, which was fabricated using computer numerical control (CNC) machining.

The Persona TASP Top Right CR 3-11/EF, Persona TASP Right EF + 0 Bottom, and the Persona Cemented Tibial Sizing Plate Size F Right were all compatible with the sensor and can be used in the same manner that the final tibial insert was used. Moreover, this technology was easily adaptable to other implant systems, and a more general design has also been created in the same manner.

This transducer used the slits to propagate the load to the strain gauges and artificial intelligence (AI) to increase the accuracy and precision of the force measurements and provide surgeons with a better tool for intraoperative load measuring, which can be seen in Figure 1.

### 2.2. Electronics

The sensors chosen were linear strain gauges since the sensing range was larger than other sensors and could support both dynamic and static loads. Three linear uniaxial 350 Ohm (SGT-1/350-TY11) precision strain gauges by Omega Engineering, Inc., Manchester, UK were used s in each compartment. The use of three was necessary for providing the AI with enough data based on the size and shape of the tibial insert surface.

The strain gauges formed part of the Wheatstone bridges along with another set of three strain gauges, which were attached to the same material to balance the bridge when unloaded and to provide temperature compensation. In total, 12 strain gauges were used for this device (6 active gauges attached to the tibial insert and 6 passive gauges used for temperature compensation) to create six half-bridge Wheatstone bridges. The sensor and strain gauges can be seen in Figure 2. Since there was an absence of a closed-formed relationship between the change in output voltage and the load and location applied, AI was needed.

The Wheatstone bridges were connected to load cell amplifiers (HX711-SEN-13879) by Sparkfun distributed by Cool Components in London, UK and then a microcontroller (Teensy 4.1 Microcontroller) on a printed circuit board (PCB) by PJRC distributed by Cool Components in London, UK. In Figure 2A,B the active gauges and the dummy gauges were paired with each other according to the numbers written next to them.

### 2.3. Artificial Intelligence (AI)

There was no closed form or linear solution to relate the applied load and location to the readings from the circuit due to the complex geometry of the sensor. As a result, AI was employed to bridge this gap, and when applied properly, it is a perfect tool for a variety of pattern recognition problems. Moreover, AI allows for the sensing area to be increased over the surface of the sensor. Figure 3A depicts the typical location of the sensors in one compartment. Figure 3B depicts the sensing area using triangulation where loads outside of this area were not accurate. Figure 3C shows the increase in the sensing area when AI is used to train the system.

An artificial neural network (ANN) is a subset of AI that classifies patterns and predicts outputs based on inputs, which was why it was chosen for this sensor. ANNs contain nodes, which have an input layer, hidden layer, and output layer. A combination of weights and biases was added to the system, and then, if the threshold was met, the inputs (with weights and biases added) were passed to the activation function and then the output (Figure 4). The advantage of using ANNs was its ability to create non-linear relationships between input and output data based on available data. This means that the sensor can predict the load and location outside of the sensing area and generalise well, which allows the sensor to respond well to real-time data.

Collecting Training Data

In order to use this network in real time, it must be trained. A training dataset must be fed to the network to determine the correct weights in biases for future prediction with real-time data. To collect this training data, data from each compartment (medial and lateral) were collected separately along with the load and location information. This was carried out to increase the chances of reaching a global minimum by allowing for fine-tuning of the parameters specific to each network. In total, four ANNs were used and implemented simultaneously to run this sensor.

To apply the loads in an organized manner, a coordinate grid system was made from 5 mm grids and aligned with the sensor for easy removal and reapplication. The training points were added based on the available space on the surface of the sensor, where the origin was based on where the farthest point to the right lies on the X-axis. The medial side had a total grid size of 3 × 6 with 18 points, and the lateral side had a total grid size of 3 × 6 as well but with 17 training points depicted in Figure 5.

The physical load sensor was made to fit the Persona Knee System developed by Zimmer Biomet in size E/F for the right knee, as seen in Figure 6. The sensor replaced the top part (No. 1 in Figure 6), which was the polyethylene insert that connects with the femoral implant once replaced, as seen in Figure 7A, as well. The Zimmer Specific sensor, Persona No. 3 (Figure 7B), the Shim 0 mm No. 2 (Figure 7C), and the tibial tray No. 4 (Figure 7D) were used together to mimic the configuration of the actual implanted knee system (Figure 8A–C). This was carried out to create congruency between training and the real-time use of the device.

To generate the training data for the ANN, a range of loads were applied to different locations. This was carried out using a Universal Testing Machine (UTM) by Testometric Micro 350/719 (Rochdale, UK). Compressive loads were applied to the surface using a 7.17 mm ball bearing (Figure 9). At each point on the Cartesian-coordinate grid, a series of loads, 49 N, 147 N, 98 N, and 196 N (5 kg, 15 kg, 10 kg, and 20 kg, respectively), were sequentially applied to each point on the surface. These loads were chosen to provide enough points to extrapolate more loads while providing a large enough range for this application, the passive loads in the knee, while still adhering to the hardware limitations due to the highly concentrated point loads. The resultant voltage readings from the unbalanced Wheatstone bridge were collected and stored for processing and training of the ANN. The training dataset was constructed in this manner to map the behaviour of the sensors at each point, then the network learns and generalises this behaviour when the contact point is larger, finding the centre of pressure using this previously created “map” of loads and locations.

2.Pre-processing Collected Training Data

The collected data were pre-processed to create the inputs and outputs for the ANN. The load and location predictions were trained in separate ANNs; however, for both, the inputs included the changes in voltages from the three strain gauges of the unbalanced Wheatstone bridges. Higher loads were extrapolated by creating a best-fit curve and added to the training dataset; then, all the values were normalised between the range of [0, 1] to allow all the values to fit on the same scale. Finally, synthetic noise was added by randomly adding noise in the range of [−5%, 5%] to the voltage reading, which aimed to help the network generalise better. After pre-processing had occurred, the next step was to train the networks, which involved tuning the parameters. Optimising these parameters was carried out by visualising the regression plots and the mean squared error to understand how the network performed on the testing dataset. Each network required investigation into the optimal network parameters. For example, the number of hidden layers and the training algorithm were variable parameters which were tuned to optimise the ANN. For the location networks, the outputs were the (X, Y) coordinates based on a Cartesian grid system created for each compartment. For the load ANN, the output was the weight in kgs.

3.Optimised Network Parameters

After the pre-processing, the training data were fed to the ANN, wherein the Bayesian regulation (BR) algorithm was used, with 85% of the data being used for training and 15% for testing. There was no validation set for this algorithm since it uses its own calculation of the mean square error for validation. The number of hidden layers used was 5 for the load networks and 10 for the location networks; this was optimised based on trial and error and the general methodology described by the authors in a previous publication for the same application [24].

In summation, the database for training was created by applying loads to different locations on the surface of the sensor, pre-processing the data, and feeding it to an ANN using the Bayesian regulation (BR) algorithm and an 85%-to-15% data split between the training and testing set. The inputs (for each compartment) to the network were the change in voltages, and the output for one network was the load, and the other network had two outputs to determine the location, which were the X and Y coordinates.

### 2.4. Aims

The aims of this paper were to investigate the accuracy and precision of the ANN in predicting the load and location on the sensor:

Testing the location:Points outside sensing area;Points not inputted into the ANN;Difference in medial/lateral sides.

Testing the load:Loads outside of the training load range;Loads not inputted to the ANN;Difference in medial/lateral sides.

## 3. Methodology

The loads and locations chosen for testing the sensor’s performance were selected to evaluate different aims of this sensor. This included the impact of using AI on the performance of the sensor when contact points were inside the triangle sensing region versus outside, the performance of one compartment compared to the other, and the performance of the sensor when the loads applied were inside versus outside of the training load range.

To test the sensor, the same UTM was used to apply a known load in Newtons (N) to the sensor at various locations on each compartment of the sensor. A larger ball bearing was used for testing the sensor where the ball bearing was used for collecting the training data (7.17 mm) and for testing (19.08 mm), which was seen in Figure 9 to understand the network’s ability to generalise with different contact points. The load was applied for 5 s before the value was recorded to allow for fluctuations; this was repeated 10 times for each load and location. Additionally, static loads were used since this was the expected use of the sensor.

Using a larger ball bearing was thought to more closely mimic the load applied intraoperatively by the femoral component. Since the training and testing points were far from the sensors, the contact point did not impact the deformation experienced by the strain gauges. This was demonstrated by Figure 10, where when the forces were the same, so were R1 and R2 for both ball bearing sizes.

To test the performance of the sensor, the loads applied to the sensor were 29 N (3 kg), 128 N (13 kg), and 226 N (23 kg). This included loads both within and outside of the training dataset. Choosing these loads provided loads that have never been applied to the sensor and are outside of the training range (for both compartments) to understand the ability of the network to predict new loads at new locations. These loads were applied to five different locations on each compartment in order to cover the whole surface of the sensor. Since each load was applied at 10 different locations over the surface of the sensor, this provided a good number of loads to investigate the aims. These testing points can be seen in Figure 11A and are described in Table 1. Moreover, Figure 11B depicts the chosen points in relation to the sensing area, where the red triangles indicate the sensing area of the sensor using triangulation.

In order to evaluate the performance of the load and location predictions, the difference between the predicted and actual values was evaluated for each of the aims. The accuracy was described as the systematic error of the average differences between the 10 trials; the precision was described as the standard deviation of the differences between the 10 trials; and the mean squared error (MSE) was described as the average of the square of the differences between the actual and estimated values. For all results, the statistical level of significance was set to *p* = 0.05.

## 4. Results

The results were investigated based on the ability of the AI to predict the load and location of testing points for a number of aims.

### 4.1. Load Predictions

The total average accuracy for the sensor was 83.41% in predicting the load and 84.63% in predicting the location. This accuracy includes the whole surface of the tibial insert and loads, which were not included in the training of the network. The following included a comparison of the load predictions by (1) compartment, (2) sensing area, and (3) training range.

Compartments

The medial and lateral load predictions were separated and compared. The actual loads applied were 3 kg, 13 kg, and 23 kg, as seen in Figure 12. The average load predictions for the medial compartment (R^2^ = 0.996) and the lateral compartment (R^2^ = 0.992) were depicted in Figure 12 as well. Both medial and lateral compartments were comparable to the actual load values applied.

2.Sensing Area

One of the aims of using AI was to increase the sensing area beyond the triangular sensing area seen in VERASENSE. Based on the curvature of the sensor and the location of the strain gauges, medial points 1, 4, and 5 and lateral points 1 and 4 were considered outside of the sensing area, and the rest were inside of the sensing area, as seen in Figure 11. Figure 13 depicts the accuracy of each load prediction inside and outside of the sensing region for both compartments.

3.
*Training Range*


To observe the impact of loads applied outside of the training dataset, the training range was considered between 5 and 25 kg. This was because, at some points in the centre of the sensor, 25 kg was added. Therefore, 3 kg was the only load that was never introduced to the ANN. So, 25 kg was neglected, and the testing load of 13 kg was classified as inside the training dataset, and 3 kg was classified as below the training dataset. Figure 14 displays the average MSE of each point, comparing the loads applied below vs. within the load training range. The average MSE for all points was almost doubled for the load applied below the training load range (3 kg).

### 4.2. Location Predictions

In Figure 15, the actual points (the same as Figure 11A) were plotted in relationship to the predicted points for the average of all the loads applied. The accuracy of the location predictions for all the loads applied was based on the distance of the prediction from the actual point and then the farthest point the network could predict and still be on the surface of the sensor.

Compartments

Figure 16 shows the average accuracy of the location predictions for each point in both the medial and lateral compartments, where there was little difference in the accuracy of the location predictions.

2.Sensing Area

The points were classified as either inside or outside of the sensing area in the same manner as the load predictions. The average results of the location predictions based on the sensing area for the three loads applied can be seen in Figure 17.

3.Training Region

The MSE of the location predictions below the training region versus within the load training region were, on average, higher below the training region (2.10) versus within (1.14), as seen in Figure 18. Compared to the load predictions, the MSE for both below and within the training region were lower for the location predictions compared to the load predictions.

## 5. Discussion

The aims of this research were to investigate the accuracy of the TKR sensor by investigating the ability of AI to simultaneously predict both the load and location of a known load. The following objectives were targeted, which included investigating (a) the sensor’s medial/lateral compartments, (b) locations inside/outside of the sensing region, and (c) the loads inside/outside of the training loading range.

The main findings were that (1) there was no significant difference between medial and lateral load and location prediction accuracies and precisions (*p* > 0.05); (2) there was no significant difference in the accuracy or precision of the load or location predictions inside versus outside the triangular sensing area (*p* > 0.05); and (3) there was a significant difference in the accuracy of load and location predictions (*p* < 0.05) when the load applied was below the training loading range. This could be because of the noise from the system causing a high signal-to-noise ratio at lower input loads.

The average accuracy across the whole sensor was 83.41% for the load predictions and 84.63% for the location predictions. The highest accuracy of 99.20% was recorded from the average of the predictions from point 2 (inside the sensing area) across the medial and lateral compartments at 13 kg, which was within the training load values. Moreover, the lower load, 3 kg (below the training load input), was worse than either the 13 kg or 23 kg load predictions (*p* < 0.05), which was demonstrated by the accuracy of the loads outside the training loading region being significantly worse than inside (*p* < 0.05) (40.57% vs. 91.15%). Therefore, when removing the 3 kg load predictions, the accuracy increases to 95.64% across the whole surface of the device. Additionally, the highest accuracy of prediction for the location was found at point 2 on the medial side and was 96.66%. Kuriyama et al. omitted location from their sensor altogether and had a relatively small sensing area; however, they were able to have a good resolution (4.45 N–0.45 kg) [25]. Reference [26] had a maximum error of 2% for their load sensor; however, their sensor was only tested with loads below 1 kg (10 N), which was not the value of the load range during TKRs. Moreover, their sensor had an operation range between 0 and 5.1 kg (0–50 N), did not sense over the whole surface, and did not display the location. Another force sensor used piezoresistive sensors to measure the load in the knee with an error of 0.51 kg (±5 N); however, the sensor only had a measurement range between 0 and 4.59 kg (0–45 N) and did not measure the location [27]. Reference [28] claimed to have an error of 0.5% in location and load prediction in a laboratory setting; however, when using a bone model, the error jumped to 13%, which may be because of the flat tibial tray design. Considering the large load sensing range of the sensor in that research, the accuracy was acceptable.

The precision of the predictions for both load and location was good; however, the average location prediction was much more precise than the load (*p* < 0.05). The average precision for the load predictions was 2.377 kg, where the location precision was 0.418 mm. This includes areas outside of the sensing area and loads outside of the training limits. Nicolet-Petersen et al. (2018) found that when observing points across the VERASENSE through a range of loads, the precision was 1.815 kg, which was not significantly different from the precision found using this sensor [21]. Alternatively, when comparing the precision of the load predictions by point (both inside and outside of the sensing area), the average was 0.462 kg, which was better than the average of 1.213 kg (*p* < 0.05) found in a study using VERASENSE [21]. Similar to the accuracy and the precision, the MSE between the medial and lateral sides were similar for the load and location predictions (*p* > 0.05); a comparison of the load and location predictions outside and inside the sensing area also had similar MSE (*p* > 0.05). Finally, in comparing the MSE of the load and location predictions to VERASENSE, the results were was 13.167 and 15.425, respectively (*p* > 0.05), which were similar. Similar results were seen with the location predictions of the medial compartment, where this sensor had an average location MSE of 0.731, and VERASENSE had an average MSE of 0.677 (*p* > 0.05). So, in summation, the MSE values for the VERASENSE and the sensor in this research were comparable.

To increase the reproducibility of the results, a Cartesian grid system should be printed on the surface of the sensors, which would decrease errors from misalignment of the grid on the surface. Although the sensor can detect higher loads, it was implied that the accuracy would decrease based on the investigation of the loads below and within the sensing range. Therefore, the accuracy of the higher loads could be greatly improved by adjustments made to the hardware, allowing point loads up to 45 kg, which would greatly increase the accuracy of the sensor at this range.

## 6. Conclusions

Intraoperative load sensors aim to measure intercompartmental loads and provide surgeons with a quantitative tool to balance the loads. This tool can be beneficial for surgeons, patients, hospitals, the National Health Services (NHS), stakeholders, and biomedical companies. Surgeons and patients can see a reduction in early revision surgeries and better postoperative outcomes. Hospitals, the NHS, and stakeholders can benefit from better patient outcomes, lower costs in revision surgeries, and shorter hospital stays postoperatively. Moreover, biomedical companies and stakeholders can use the data of load distributions intraoperatively to better design implant systems and inform regulatory agencies.

This sensor was amended to fit Persona’s shim system and can also be adjusted to have a more general design to incorporate other implant systems. The benefit of this sensor was that its use of AI allowed it to identify loads and locations over the entire surface of the insert while also upholding high accuracy and precision over a wide range of loads. The design of the hardware and software was carried out in such a way as to improve the accuracy of intraoperative knee sensing by increasing the sensing surface and the sensing load range compared to other knee force sensors found in the literature. This research investigated the accuracy of the sensor by investigating the ability of AI to predict both the load and location of a point load, which was different from what was used for training. The sensor performed with good accuracy and precision in both contact load and location predictions. The main findings were that both medial and lateral load and location prediction accuracies were similar (*p* > 0.05), the accuracy and precision inside and outside of the sensing area were comparable (*p* > 0.05), and finally, there was a significant difference in the accuracy of load and location predictions (*p* < 0.05) when the load applied was below the training load range. Incorporating a wider range of load values into the training process could significantly enhance the results of the sensor. An added benefit of using AI is that retraining can be performed quickly and simply as more research is carried out on the performance and training process. This ensures that the sensor’s performance can evolve continuously as new data emerge.

## Figures and Tables

**Figure 1 sensors-24-00585-f001:**
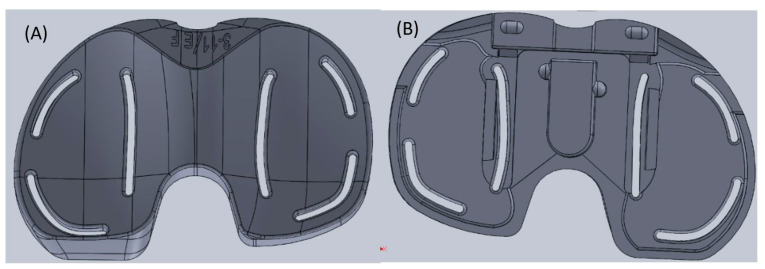
Physical sensor design: (**A**) top view; (**B**) bottom view.

**Figure 2 sensors-24-00585-f002:**
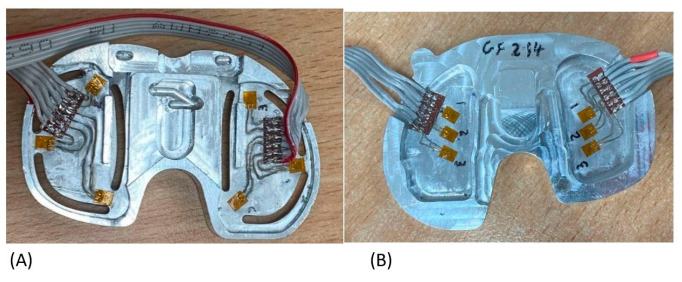
(**A**) Active sensor. (**B**) Temperature compensation.

**Figure 3 sensors-24-00585-f003:**
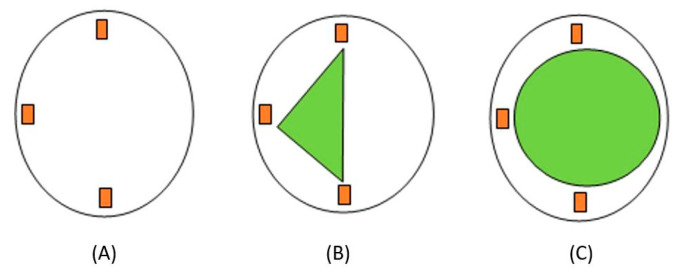
Strain gauge placement and sensing area: (**A**) strain gauge placement; (**B**) sensing area using triangulation; (**C**) sensing area using AI.

**Figure 4 sensors-24-00585-f004:**
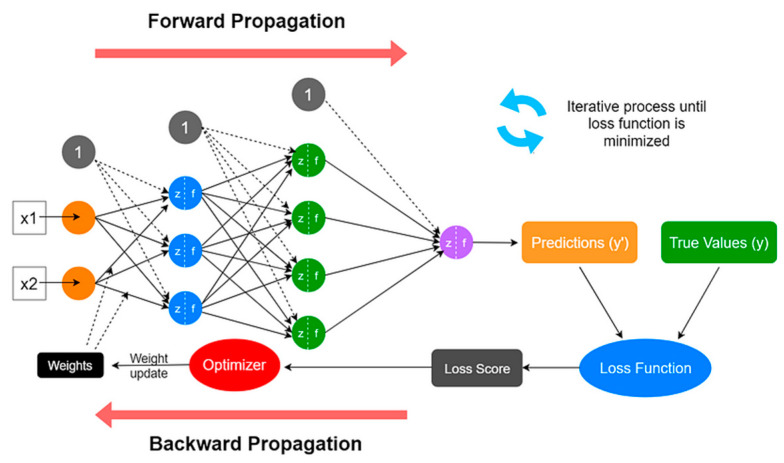
Artificial neural network structure [23].

**Figure 5 sensors-24-00585-f005:**
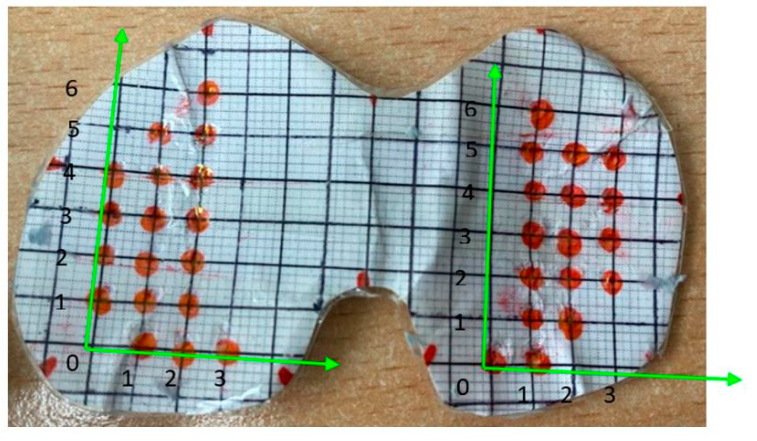
Cartesian coordinate system on the sensor’s surface.

**Figure 6 sensors-24-00585-f006:**
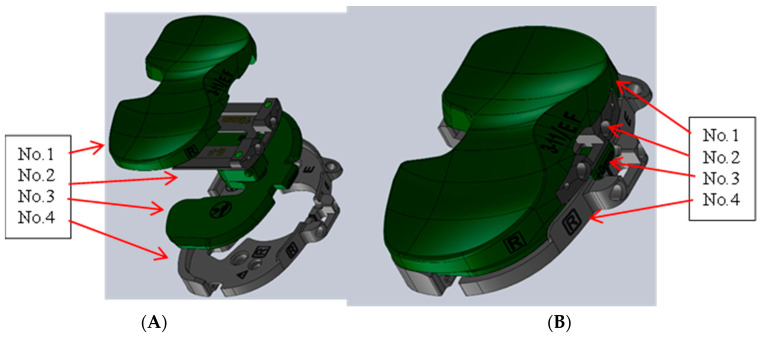
Persona Knee System CAD. (**A**) Exploded view of Persona configuration. (**B**) Persona configuration.

**Figure 7 sensors-24-00585-f007:**
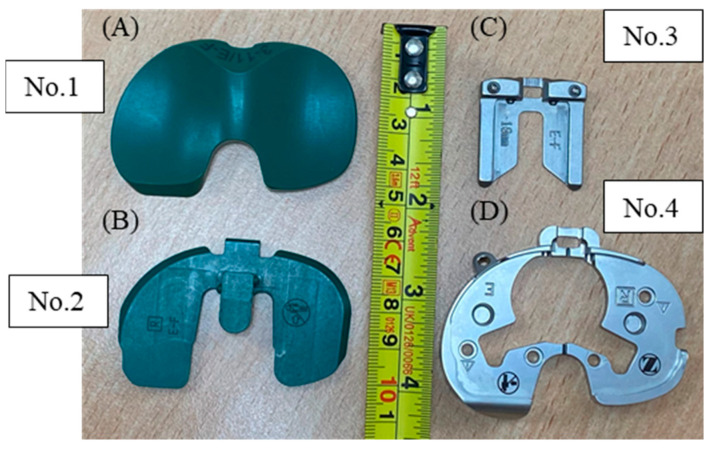
Persona Knee System. (**A**) Polyethylene insert No. 1 (top view). (**B**) Persona TASP right EF. (**C**) TASP shim. (**D**) Persona cemented tibial sizing plate right.

**Figure 8 sensors-24-00585-f008:**
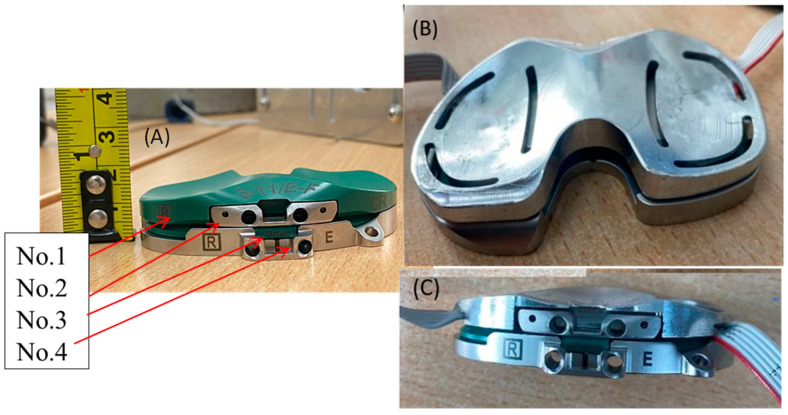
(**A**) Persona configuration. (**B**) Sensor configuration (top view). (**C**) Sensor configuration (front view) 1.

**Figure 9 sensors-24-00585-f009:**
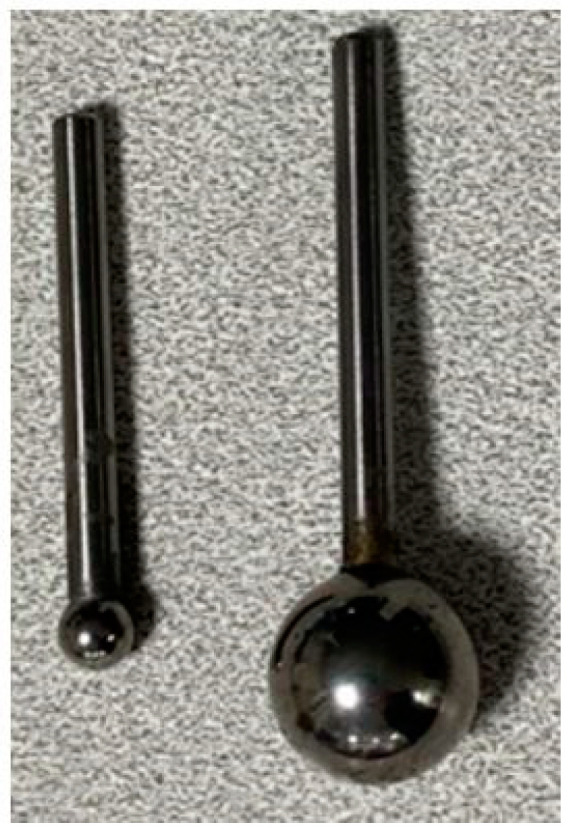
Ball bearing used for training (7.17 mm) and testing (19.08 mm).

**Figure 10 sensors-24-00585-f010:**
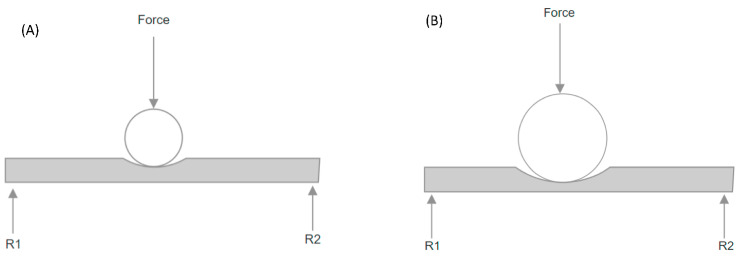
Impact of ball bearing size on reaction forces. (**A**) Smaller ball bearing; (**B**) larger ball bearing.

**Figure 11 sensors-24-00585-f011:**
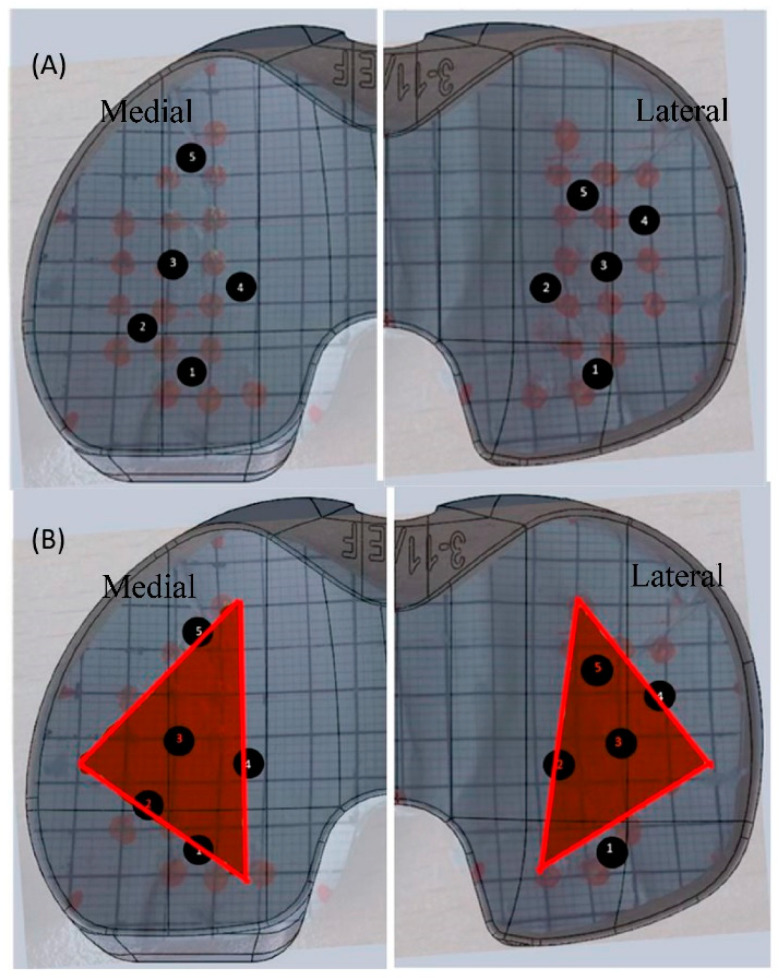
(**A**) Testing points on the surface of the sensor where the black numbers refer to the location of the applied testing load. (**B**) Testing points in relation to the sensing area where the red triangle refers to the sensing area and the black numbers as the location of the testing loads in proximity to the sensing area.

**Figure 12 sensors-24-00585-f012:**
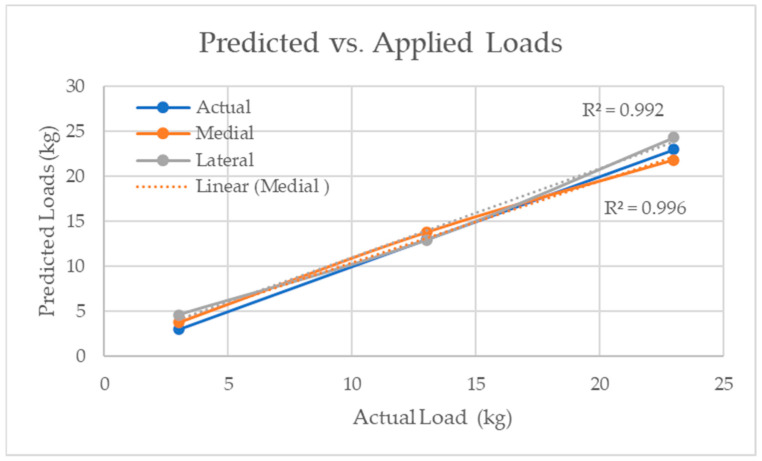
Predicted load compared to the actual applied load for both medial and lateral compartments.

**Figure 13 sensors-24-00585-f013:**
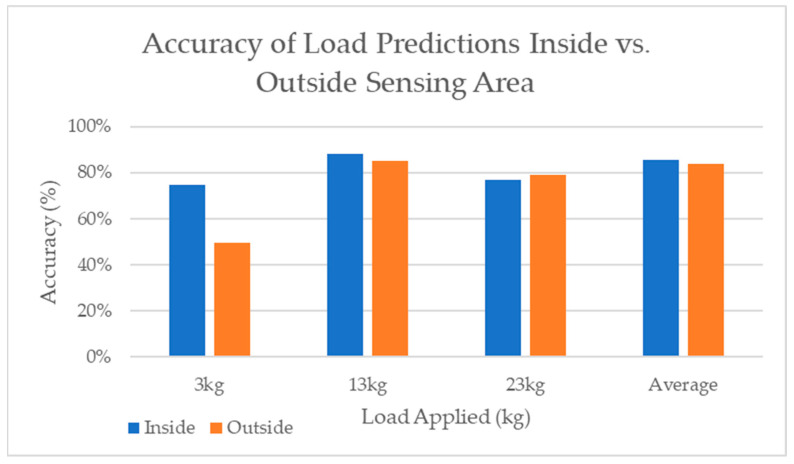
Accuracy of each load prediction inside versus outside of the sensing area.

**Figure 14 sensors-24-00585-f014:**
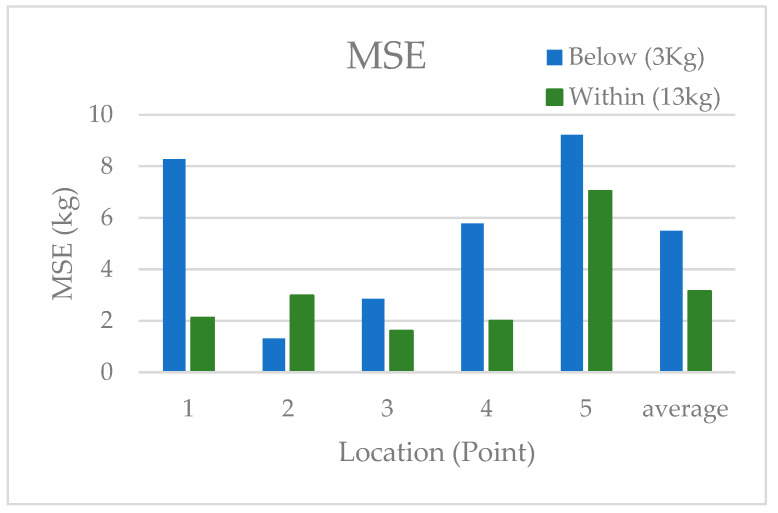
MSE at each point inside versus outside of the training load range.

**Figure 15 sensors-24-00585-f015:**
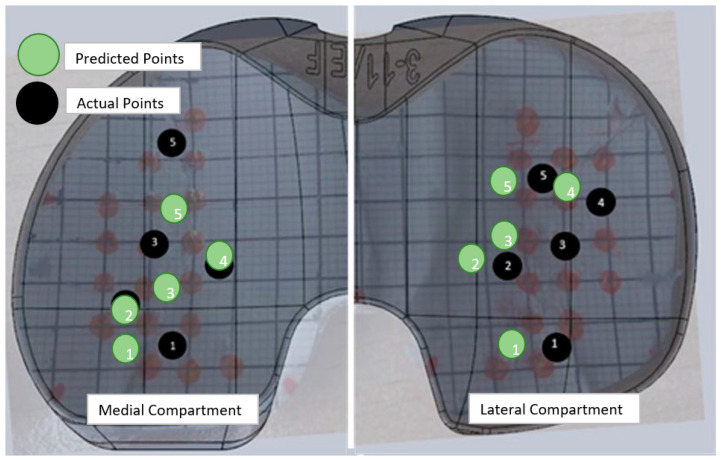
Average location predictions for medial and lateral compartments compared to the actual location of the applied load where the black numbers represent the location of the actual applied loads and the green as the predicted locations of the applied loads.

**Figure 16 sensors-24-00585-f016:**
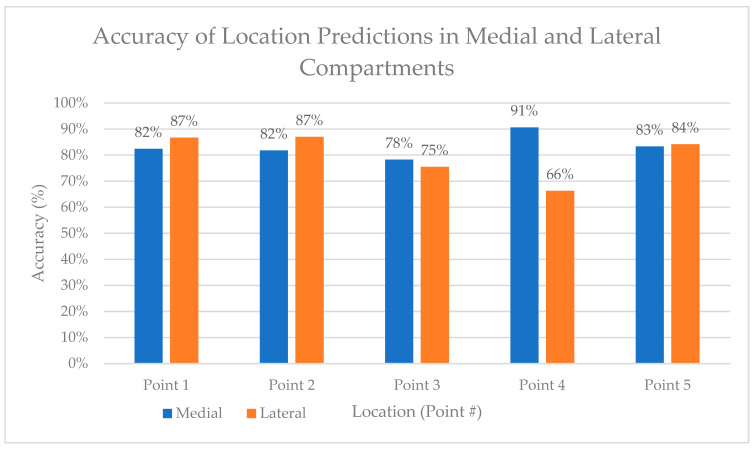
Accuracy of location predictions in medial and lateral compartments.

**Figure 17 sensors-24-00585-f017:**
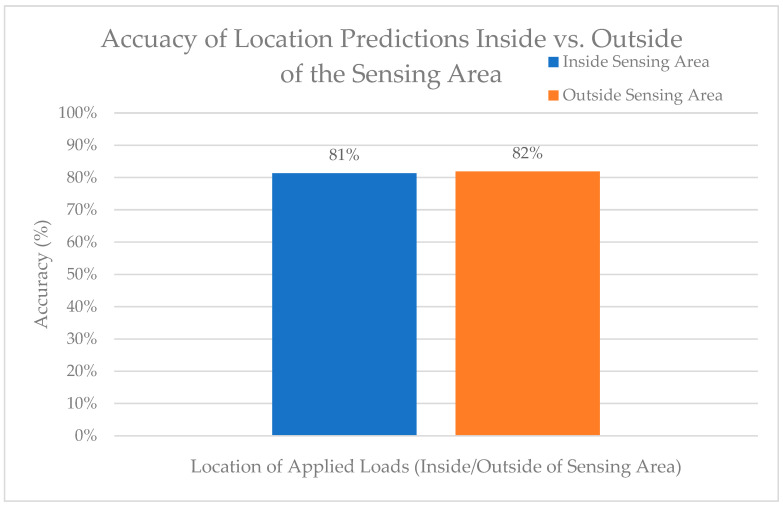
Accuracy of location predictions inside versus outside of the sensing area.

**Figure 18 sensors-24-00585-f018:**
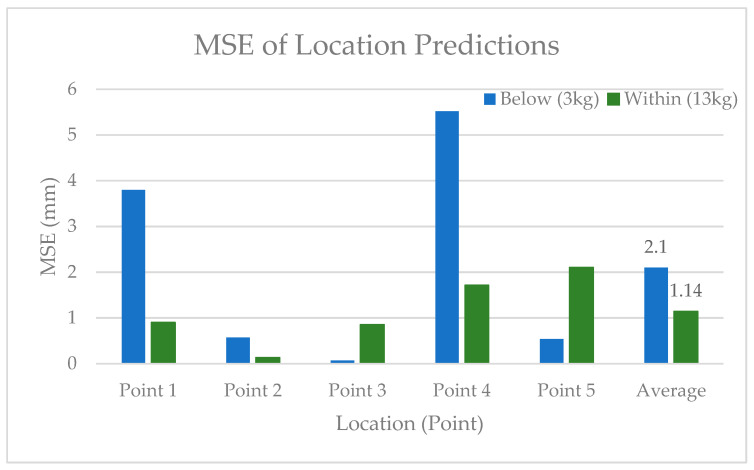
MSE of the location predictions inside and outside of the training load region.

**Table 1 sensors-24-00585-t001:** Testing points with (X, Y) coordinates.

	Medial Compartment (X, Y)	Lateral Compartment (X, Y)
Point 1	(1.5, 0.5)	(1.5, 0.5)
Point 2	(0.5, 1.5)	(0.5, 2.5)
Point 3	(1.0, 3.0)	(2.0, 3.0)
Point 4	(2.5, 2.5)	(3.0, 4.0)
Point 5	(1.5, 5.5)	(1.5, 5.5)

## Data Availability

The data presented in this study are available on request from the corresponding author.

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
