# Peer review of "Exploring the Performance of an Artificial Intelligence-Based Load Sensor for Total Knee Replacements"

_sensors, 2024, doi:10.3390/s24020585_

Round 1

Reviewer 1 Report

Comments and Suggestions for Authors

To authors:

This paper explored the possibility of using artificial neural networks (ANN), trained on experimental data, to improve the accuracy and precision of load and location predictions from load sensors used in total knee replacement (TKR). The training data were collected from a self-developed load sensors and housing structures, with known loads applied at different pre-defined locations. In total, 35 training points with 4 different loads were included. During validation, 3 different loads were tested at 5 different locations, with both of which configured to test predictions in- and outside of the training data ranges. The results suggested that the medial and lateral load and location predictions were similar in accuracy and precision, the accuracy and prediction in- and outside of the sensing area were comparable, and the accuracy of load and location predictions differ significantly when the applied load was below the training data range.

The paper is well written and easy to read - the research idea is conveyed clearly and concisely, the experiments are explained and discussed with good details. However, this reviewer has two major concerns:

1) as the core novelty of the paper, the ANN employed was not sufficiently introduced and explained. For example, what’s the converge criteria used for training? What optimizations have been done? Why training 4 ANNs separately?

2) Testing loads used during validations were too limited (only 3 loads with 1 below, 1 within, and 1 above the training range). it is also unclear why these specific loads were chosen (similar to actual loads during TKR? Hardware limitations?) Moreover, as admitted by the authors in L279, there was a human error that further limited the scope of the results in 4.1.3).

Other comments:

- Fig. 4: how are the origins of the coordinates defined? What if the size and shape of the surface changes?

- Fig. 5,6,7 are confusing: consider marking sensor location directly on Fig. 5, link/refer parts in Fig. 6 and 7 back to Fig. 5, and use simplified terms (e.g., there is no need to spell out “Personal TASP Top Right CR 3-11/EF”, especially when it’s already introduced in previous sections)

- Why testing with static loads only? Also, practically, more than one point will be loaded - why not training and predicting loads across a surface area, mimicking actual TKR procedures?

Minor comments:

- Introduction is unnecessarily focused on statistics and impacts on UK and NHS

- L43: typo “ageing”

- L321: extra space between “accuracy” and “or”

- L360: typo “thewas”

- L377: duplicated introduction of acronym “AI”

Author Response

  • as the core novelty of the paper, the ANN employed was not sufficiently introduced and explained. For example, what’s the converge criteria used for training? What optimizations have been done? Why training 4 ANNs separately?

I added about how I trained my network, ie choosing hidden layers. The reason I used 4 ANNs is because of the main goal of the sensor which is to measure the amplitude of the load and the location in each compartment. Therefore, the two compartments need to be treated as two separate systems (medial and lateral). Then we need two different estimations for each compartment, one for the measurement of the load amplitude and where its centre of pressure is located. Separating the ANNs networks increased the chances of reaching a global minimum since if I combined the load and location networks for each compartments I would have 3 inputs and 3 outputs which would decrease the chances of convergence. I also was then able to fine tune the parameters for each network which I believe made the results more accurate. However, the training of the network is described in another paper. The database for training is created by applying loads to different locations on the surface of the sensor, preprocessing the data, feeding it to an ANN using Bayesian regulation (BR) algorithm and an 85% to 15% data split between training and testing set. There is no validation set for BR. The inputs (for each compartment) were the change in voltages and the output for one network was the load amplitude and the other had two which were the X and Y coordinates. I have extensively investigated optimal parameters for this sensor but it is not relevant to this paper, I believe.

  • Testing loads used during validations were too limited (only 3 loads with 1 below, 1 within, and 1 above the training range). it is also unclear why these specific loads were chosen (similar to actual loads during TKR? Hardware limitations?) Moreover, as admitted by the authors in L279, there was a human error that further limited the scope of the results in 4.1.3).

I have attempted to add this in by stating that because each load was added 5 times to each side, this increased the number of loads below to 10 total and within to 10 which was enough to form a conclusion. Training loads were chosen based on hardware limitations. New design iterations with materials with higher yield strengths would allow for higher loads being able to be applied for training. Moreover, the human error was addressed by in the future printing the grid to the surface of the sensor.

Other comments:

- Fig. 4: how are the origins of the coordinates defined? What if the size and shape of the surface changes?

I aimed to explain that the coordinate system was arbitrarily made, as I am teaching the network that this point (X,Y), Number, Letter, whatever I choose gives me this type of behaviour of the gauges. I chose (X,Y) since it was easiest to understand and implement with my GUI. I made the origin line up with the farthest X coordinate to the left on my sensor for consistency, and so most of my predictions would be positive. Since the goal of the sensor was for knee balancing between compartments as long as the grids were made similarly it made comparing the compartments intuitive.

- Fig. 5,6,7 are confusing: consider marking sensor location directly on Fig. 5, link/refer parts in Fig. 6 and 7 back to Fig. 5, and use simplified terms (e.g., there is no need to spell out “Personal TASP Top Right CR 3-11/EF”, especially when it’s already introduced in previous sections)

Addresses

- Why testing with static loads only? Also, practically, more than one point will be loaded - why not training and predicting loads across a surface area, mimicking actual TKR procedures?

Added to paper explain that static loading is what is being measured during the TKR by the surgeons. The knee is held at certain orientation and measured so for testing we used static loads. The goal of using an ANN was to allow it to learn and generalise for different loading condition, we loaded a lot of points over the surface so it would learn to generalise and find the centre of pressure. Different implant systems have different shapes and sizes and providing training data that is too similar or specific to the one system of loading condition limits its ability to be able to accurately predict new contact point locations and loads.  

Minor comments:

- Introduction is unnecessarily focused on statistics and impacts on UK and NHS

This research is occurring in the UK therefore the focus on the UK, however same can be implied by anywhere else in the work

- L43: typo “ageing”

Ageing is British spelling, I have changed it to US

- L321: extra space between “accuracy” and “or”

Addressed

- L360: typo “thewas”

Addressed

- L377: duplicated introduction of acronym “AI”

Addressed

Reviewer 2 Report

Comments and Suggestions for Authors

Revision of Exploring the Performance of an Artificial Intelligence-Based 2 Load Sensor for Total Knee Replacements

The presented work is interesting and proposes an application of AI for the prediction of a load sensor for total knee replacement.

The activities have been described even if I found some points that should be integrated. Moreover, some results should be clarified, before accepting this work for publication.

Here in the following my questions:

1)      The paper focuses on the application of AI for predicting the load the sensor should measure out of the sensing area as well as for predicting the load location. However, it seems to me that the architecture of the neural network has not been fully described: how is the database? How much data have been used to train the network, how much to validate, and how much to test? Which functions have been applied? What are the input parameters? Some information is reported in paragraph 2.3, but I believe this part should be improved with quantitative data.

2)      I did not understand why two spherical indenters have been used. I understood the motivation for using a bigger indenter, thus I do not understand why using a smaller one, since then the applied force is the same.

3)      Figure 12: Please, note that you report “inside vs outside” but then the first bar is “outside”, and the second is “inside”; moreover, in the following graphs the blue colour is associated with “inside”, while orange is “outside”. Thus, I suggest inverting the colors and legend of the figure.

4)      Figure 15, how were these accuracies calculated? For example, point 5 in the medial compartment seems to be quite far from the real location, but accuracy reports 83%. On the contrary, point 3 (same compartment) seems more close to real point 3, even if the accuracy is 78%. Points 1 and 2 have the same accuracy, but they seem quite different. How are these numbers determined? For which applied load?

5)      Discussions are missing the limitations of the work and how these limits have been overcome or accepted with motivation.

6)      I am not sure I have understood how the sensor commonly works: in the conclusion, I would stress the improvements this research would add to the actual sensor.

Minor suggestions:

1)      Affiliations are missing;

2)      Abstract lines 7-9: “Having a sensor that is accurate, precise, can sense over the whole surface, and can include a wide range of loads is important to the success of a sensor”, the sentence does not sense and the word “sensor” is repeated.

3)      Paragraph 2.3: “Under the umbrella of AI exists neural networks which aim to mimic the neurons in the 150 brain. Artificial Neural Networks (ANNs) can classify patterns and predict outputs based 151 on inputs, which was why it was chosen for this sensor.” I believe this sentence should be reformulated.

4)      Figure 10 appears cited before figures 8 and 9. Maybe you should consider unifying some figures into a few bigger ones.

5)      I would improve the graphics of the figures: some are formed by pictures of different sizes, and text styles (Times New Roman or Arial or maybe others)…

6)      Maybe this recent review could be of interest when introducing the topic of OA and the necessity of total knee replacement: DOI: 10.3390/pr11041014

Comments on the Quality of English Language

Some sentences should be reformulated

Author Response

  • The paper focuses on the application of AI for predicting the load the sensor should measure out of the sensing area as well as for predicting the load location. However, it seems to me that the architecture of the neural network has not been fully described: how is the database? How much data have been used to train the network, how much to validate, and how much to test? Which functions have been applied? What are the input parameters? Some information is reported in paragraph 2.3, but I believe this part should be improved with quantitative data.

Added notes about how I trained my network, ie preprocessing and choosing network parameters. However, the training of the network is described in another paper. The database for training is created by applying loads to different locations on the surface of the sensor, preprocessing the data, feeding it to an ANN using Bayesian regulation (BR) algorithm and an 85% to 15% data split between training and testing set. There is no validation set for BR. The inputs (for each compartment) were the change in voltages and the output for one network was the load and the other had two which were the X and Y coordinates. I have extensively investigated optimal parameters for this sensors but it is not relevant to this paper, I believe.

  • I did not understand why two spherical indenters have been used. I understood the motivation for using a bigger indenter, thus I do not understand why using a smaller one, since then the applied force is the same.

I have added to the paper to explain this but using a small ball bearing for training was to be able to get close to the gauges and to provide higher resolution for the user ie a finer grid size while still have reliable data and then for testing I wanted to investigate the networks ability to generalise for different contact shapes as different implant systems use different shapes and sizes of the femoral component.

  • Figure 12: Please, note that you report “inside vs outside” but then the first bar is “outside”, and the second is “inside”; moreover, in the following graphs the blue colour is associated with “inside”, while orange is “outside”. Thus, I suggest inverting the colors and legend of the figure.

Addressed

  • Figure 15, how were these accuracies calculated? For example, point 5 in the medial compartment seems to be quite far from the real location, but accuracy reports 83%. On the contrary, point 3 (same compartment) seems more close to real point 3, even if the accuracy is 78%. Points 1 and 2 have the same accuracy, but they seem quite different. How are these numbers determined? For which applied load?

I added to the paper to explain this but the accuracy for the location predictions was based on the farthest distance the network could have possibly predicted while remaining on the surface of the sensor vs. how far the point was predicted from the actual point.

  • Discussions are missing the limitations of the work and how these limits have been overcome or accepted with motivation.

Addressed, adding a point about printing the coordinate system on the surface of the sensor for reproducible training and testing.

  • I am not sure I have understood how the sensor commonly works: in the conclusion, I would stress the improvements this research would add to the actual sensor.

I have now added that this sensor can measure loads over the whole surface (a limitation of other sensors) and remains accurate. Also, it integrates the complicated shim design of the Persona Knee system for easy adjustment of thickness.

Minor suggestions:

  • Affiliations are missing;

Addressed with editor

  • Abstract lines 7-9: “Having a sensor that is accurate, precise, can sense over the whole surface, and can include a wide range of loads is important to the success of a sensor”, the sentence does not sense and the word “sensor” is repeated.

Addressed

  • Paragraph 2.3: “Under the umbrella of AI exists neural networks which aim to mimic the neurons in the 150 brain. Artificial Neural Networks (ANNs) can classify patterns and predict outputs based 151 on inputs, which was why it was chosen for this sensor.” I believe this sentence should be reformulated.

Addressed

  • Figure 10 appears cited before figures 8 and 9. Maybe you should consider unifying some figures into a few bigger ones.

Figure was misnamed- Addressed

5)      I would improve the graphics of the figures: some are formed by pictures of different sizes, and text styles (Times New Roman or Arial or maybe others)…

 Changed to times new roman- Addressed

6)      Maybe this recent review could be of interest when introducing the topic of OA and the necessity of total knee replacement: DOI: 10.3390/pr11041014

Implemented and added more recent papers too

Reviewer 3 Report

Comments and Suggestions for Authors

1- The summary needs to include a presentation of the researcher's findings and a comparison of the researcher's results with those of other researchers.

2- The introduction needs to be updated with recent sources.

3- The researcher could have enhanced their study by examining stress distributions, calculating the safety factor, and additionally modeling the replaced knee using modern modeling and analysis techniques.

4- The results require a detailed discussion as the existing discussion is very concise.

5- The conclusions need to be written in a way that focuses on presenting conclusions corresponding to each goal of the study.

Author Response

- The summary needs to include a presentation of the researcher's findings and a comparison of the researcher's results with those of other researchers.

Because there has not been a sensor made that uses AI for knee force tracking direct comparisons can be not be made. I have compared the accuracy of various other intraoperative knee sensors, including commercial sensors like VERASENSE and independent load sensors made by various other authors.  

2- The introduction needs to be updated with recent sources.

 Added two sources from 2023

3- The researcher could have enhanced their study by examining stress distributions, calculating the safety factor, and additionally modeling the replaced knee using modern modeling and analysis techniques.

This was outside of the scope of this study, I have another paper which focuses on this.

4- The results require a detailed discussion as the existing discussion is very concise.

Added to discussion

5- The conclusions need to be written in a way that focuses on presenting conclusions corresponding to each goal of the study.

Added to conclusion

Round 2

Reviewer 2 Report

Comments and Suggestions for Authors

I thank the authors for their improvements in the work.

However, referring to my first point (details about the ANN), I understand this is not a paper focused on the network and its architecture. Still, I think the details you reported in the answer should be added in the manuscript too ("The database for training is created by applying loads to different locations on the surface of the sensor, preprocessing the data, feeding it to an ANN using Bayesian regulation (BR) algorithm and an 85% to 15% data split between training and testing set. There is no validation set for BR. The inputs (for each compartment) were the change in voltages and the output for one network was the load and the other had two which were the X and Y coordinates.")

Moreover, it seems that pictures still have some differences in fonts, format etc...

Author Response

Thank you for taking the time to review this paper. I have added a small paragraph about the training and fixed the figures.